# Transcriptome-Assisted SNP Marker Discovery for *Phytophthora infestans* Resistance in *Solanum lycopersicum* L.

**DOI:** 10.3390/ijms24076798

**Published:** 2023-04-05

**Authors:** Saptarathi Deb, Maria Cristina Della Lucia, Samathmika Ravi, Giovanni Bertoldo, Piergiorgio Stevanato

**Affiliations:** Department of Agronomy, Food, Natural Resources, Animals and Environment (DAFNAE), University of Padova, 35020 Legnaro, PD, Italy

**Keywords:** late blight, *Phytophthora infestans*, tomato, transcriptome, SNP-markers

## Abstract

Late blight, caused by oomycetes *Phytophthora infestans* is one of the most challenging fungal diseases to manage in tomato plants (*Solanum lycopersicum* L.). Toward managing the disease, conventional breeding has successfully introgressed genetic loci conferring disease resistance from various wild relatives of tomato into commercial varieties. The cataloging of disease-associated SNP markers and a deeper understanding of disease-resistance mechanisms are needed to keep up with the demand for commercial varieties resistant against emerging pathogen strains. To this end, we performed transcriptome sequencing to evaluate the gene expression dynamics of tomato varieties, resistant and susceptible to *Phytophthora* infection. Further integrating the transcriptome dataset with large-scale public genomic data of varieties with known disease phenotypes, a panel of single nucleotide polymorphism (SNP) markers correlated with disease resistance was identified. These SNPs were then validated on 31 lines with contrasting phenotypes for late blight. The identified SNPs are located on genes coding for a putative cysteine-rich transmembrane module (CYSTM), Solyc09g098310, and a nucleotide-binding site–leucine-rich repeat protein, Solyc09g098100, close to the well-studied *Ph-3* resistance locus known to have a role in plant immunity against fungal infections. The panel of SNPs generated by this study using transcriptome sequencing showing correlation with disease resistance across a broad set of plant material can be used as markers for molecular screening in tomato breeding.

## 1. Introduction

*Solanum lycopersicum* L., one of the most consumed fruit, has emerged to be an integral part of the global diet in recent centuries [1]. Oomycetes *Phytophthora infestans*, a fungal pathogen causing the disease late blight, is amongst the most difficult diseases to manage in tomato plants [2]. The disease progression and recent epidemic outbreaks have impacted cultivation severely throughout the globe resulting in huge economic losses [3]. Moreover, recent climate change events have increased the incidence of more aggressive fungal attacks affecting tomato yield and quality [2]. The infection is associated with symptoms that include dark lesions in the foliage, fruit, and stem followed by chlorotic spots and necrosis [2].

Over the decades, strategies to manage the disease have majorly focused on the use of fungicides and selective resistance breeding [2]. More recently, research efforts in transgenics and genome editing have resulted in the alteration of expression of resistance genes and their regulators [4,5]. However, with the ongoing debate and ethical concerns regarding genome editing, conventional selective breeding is the most acceptable tool at our disposal to control the disease. Currently, the introgression of resistance genes and quantitative trait loci (QTLs) from wild *Solanum* relatives into commercial tomato varieties has been the major source of resistance. The most prominent of these resistant genes (R gene) loci are *Ph-1,2,3,5-1,5-2* from *Solanum pimpinellifolium* [6,7] and *Ph-4,3* from *Solanum habrochaites* [8,9]. These resistant factors individually or in combination have been able to provide partial-to-broad resistance to various *Phytophthora* species. A combination of *Ph-2* and *Ph-3* has been one of the most successful introgressions into breeding and commercial lines like NC1 CELBR, NC2 CELBR, Mountain Magic, and Mountain Merit that have resulted [10]. Regardless of the efforts made so far to achieve resistance, there is an incessant need for monitoring these host-pathogen interactions through other novel genes due to the resistance breakdown of emerging fungal isolates [10].

Another direction of conferring resistance to late blight is by developing strategies to exploit the use of microRNAs (miRNAs). Previous studies by Luan et al. [4,11] demonstrated the involvement of many miRNAs including miR172 and miR482 in disease resistance or susceptibility [4,12]. Using degradome data, it was observed that many R genes serve as the primary target of these miRNAs [12]. Along with R genes, some of these miRNAs were also involved in perturbing the expression of genes involved in scavenging reactive oxygen species (ROS), hence lowering the physiological damage caused by ROS upon infection and thereby improving the fitness of these plants [12].

To achieve such molecular breeding strategies and track the resistance loci, single nucleotide polymorphism (SNP) markers have proven to be extremely helpful [13]. With advances in next-generation sequencing technologies and growing genomic resources, QTL discovery and the development of SNP markers has become more significant and efficient than previous strategies like Simple Sequence Repeat (SSR) markers and Cleaved Amplified Polymorphic Sequences (CAPS) markers [14]. In the context of SNP markers for late blight resistance in tomato plants, a study by Arafa et al. [13] reported SNP markers identified using a genome-wide association study (GWAS) for resistance against the *Phytophthora* Egyptian isolate. While resistance-associated molecular markers have been reported [13,15], there is a need for more such molecular markers as the source of resistance can be quantitative, and thereby confer a broader scope of resistance that can be used against different isolates of the fungus.

Along with the discovery of markers for breeding, it is also important to understand the disease dynamics and the host response in different resistant and susceptible lines. In this context, the aim of this study was to identify SNP markers in functional genes associated with disease resistance and validate them across a range of plant material. This was primarily achieved using an RNA sequencing-guided approach to understand the transcriptomic changes upon infection by *Phytophthora infestans* in resistant and susceptible breeding lines, followed by the discovery of SNPs in disease-responsive genes dysregulated upon infection between resistant and susceptible lines. The selected SNP markers were additionally correlated primarily with all publicly available sequencing datasets of tomato plants with contrasting disease phenotypes and finally validated on a test dataset of secondary lines with known disease phenotypes. Thirty-nine candidate SNPs discriminating the sequenced resistant and susceptible lines in our dataset were screened across publicly deposited lines with known phenotypes and eight SNPs from two genes were finally validated by correlating with disease phenotype on a broader germplasm. The set of validated SNP assays generated by this study can serve as a valuable resource for the tomato breeding community.

## 2. Results

### 2.1. Transcriptome and Differential Gene Expression Analysis

A total of 100,187,608 reads coming from transcriptome sequencing of eight samples showed an average mapping percentage of 82.83% to the SL3.0 genome. Interestingly, the unmapped reads from each sample when mapped to the *Phytophthora infestans* genome showed the presence of *Phytophthora infestans* only in infected susceptible plants (Appendix A). This indicated an active progression of the disease in the susceptible infected plants compared to resistant and control plants at 48 h post-infection.

PCA analysis performed from a read counts matrix showed a distinct cluster of infected and control plants in both resistant and susceptible varieties. The clustering further strengthens our confidence in the plant material and uniformity of replicates in terms of their gene expression patterns and differences between conditions (Figure 1a)

Differential gene expression analysis between conditions resulted in several differentially expressed genes, shown in Figure 1b, between various comparisons detailed below.

k-means clustering of differentially expressed genes with k = 4 (Figure 1c) followed by an enrichment analysis showed two clusters of interest: (i) Cluster D showed a remarkably similar pattern of upregulation in control resistant and infected susceptible individuals. This cluster comprised the genes involved in major immune system pathways including jasmonic acid and other defense responses. Meanwhile, (ii) Cluster A showed a pattern of upregulation in infected resistant and control susceptible individuals. These genes were involved in plant growth and development.

The gene clustering results indicate the tendency of the resistant plants to have an active defense even before the onset of infection, whereas the susceptible plants show an immune system activation only after the onset of infection. 

Following an overview of gene expression analysis, we focused on two main comparisons.

The comparison of control susceptible vs control resistant showed a total of 2997 differentially expressed genes—1749 downregulated and 1228 upregulated—in control susceptible individuals. The downregulated genes were annotated as involved in the regulation of cellular metabolic processes, response to stress, regulation of macromolecule metabolic processes, regulation of primary metabolic processes, and cellular response to stimuli. The upregulated genes were involved in the regulation of macromolecule metabolic processes, regulation of gene expression, regulation of cellular metabolic processes, transport, and establishment of localization.A comparison of susceptible infected vs resistant infected showed a total of 3473 differentially expressed genes—1465 downregulated and 2007 upregulated—in susceptible infected individuals. The downregulated genes were annotated to be involved in the regulation of cellular metabolic processes, regulation of macromolecule metabolic processes, regulation of primary metabolic processes, nucleobase-containing compound biosynthetic processes, and regulation of nitrogen compound metabolic processes. The upregulated genes were involved in the establishment of localization, transport, phosphate-containing compound metabolic processes, phosphorus metabolic processes, and regulation of cellular metabolic processes.

### 2.2. SNP Analysis from Transcriptome Data

The total number of variants commonly present in resistant and susceptible individuals was 25,857 and 19,200, respectively. Upon screening the differentially expressed genes in all conditions of comparison, 29 variants from the resistant lines belonging to 10 differentially expressed genes were identified (Appendix A). By screening known late blight-resistant proteins from the literature [16], a total of 10 missense variants were found in Solyc09g098100, a nucleotide-binding site–leucine-rich repeat protein annotated as topovirus resistance protein C by ITAG3.0. These variants were further checked in publicly available datasets, and the confident targets were taken for experimental validation (Appendix A).

### 2.3. Integration of Public Data for Confidence 

The same 39 variants identified above were screened in 17 tomato varieties (13 resistant and 4 susceptible to *Phytophthora infestans*) consisting of *Solanum lycopersicum* and *Solanum pimpinellifolium.* Interestingly, out of 39 identified SNP variants, 16 SNPs were found to be present in more than two resistant varieties and absent in all susceptible varieties. This analysis resulted in a confident set of 16 SNPs for large-scale experimental validation. (Appendix A). 

### 2.4. Genotyping-Based SNP Validation 

The identified set of 16 SNP variants from transcriptomics and public data analysis was validated by screening a wide range of 14 resistant and 17 susceptible lines. SNPs with a status of a homozygous resistant allele in the susceptible plants were rejected and finally, a total of seven SNPs were found to be associated with the resistant varieties. The association of these alleles with the resistance phenotype was further tested using a chi-squared test of independence. The summary of the allele frequencies and their chi-squared statistical values are presented in Figure 2 and Table 1.

The validated SNPs included one missense mutation from Solyc09g098310 and seven missense mutations from Solyc09g098100. The detailed SNP information with sequences is provided in Appendix A.

### 2.5. Characterization of Candidate Genes

Annotation of the two genes containing the validated SNPs (Solyc09g098310, Solyc09g098100), identified Solyc09g098310 as putative a cysteine-rich transmembrane (CYSTM1) family protein and Solyc09g098100 as topovirus resistance protein C with maximum identity followed by a putative late blight resistance protein homolog R1A-3. Domain analysis using InterProScan showed the presence of a cysteine-rich and transmembrane domain in Solyc09g098310 and a nucleotide-binding domain and a leucine-rich repeat domain (NLB-LRR) domain in Solyc09g098100 (Figure 3b). The analysis of cis-acting regulatory sites showed the presence of a salicylic response element, which is known to have a role in fungal infection response [17]. miRNA-target analysis identified the presence of sly-miR395a/b in Solyc09g098310 transcript whereas Solyc09g098100 mRNA transcript had multiple miRNA binding sites with top hits including sly-miR482, sly-miR5303, sly-miR396, sly-miR6027, sly-miR156, sly-miR396, and sly-miR6024.

## 3. Discussion

In any host-pathogen study, understanding the genetic basis of resistance and tracking them in the population has been a fundamental strategy. Considering that tomato breeding is extremely fast-paced with a minimum of two to three growing cycles per year, the monitoring of host defense and pathogen evolution through molecular markers becomes even more pertinent for resulting yield and trait improvements. Furthermore, with the increasing pressure from aggressive pathogen strains, there is a huge demand for molecular markers identified in plants able to resist current pathogen strains [18].

In this study, we used RNA sequencing to understand the transcriptomic differences between resistant and susceptible tomato lines upon infection with *Phytophthora infestans.* The major changes indicated a shift by upregulation of immune response pathways in susceptible plants upon infection, whereas resistant plants had an overall active immune response even during pre-infection as represented by cluster D, Figure 1c. 

Previous studies to identify SNP markers for late blight resistance in tomatoes have employed a genome-wide association study (GWAS) with DNA sequencing data including whole genome resequencing (WGRS) and reduced genome sequencing techniques like ddRAD seq [13]. As RNA sequencing directly queries the functional part of the genome (mRNA), this allowed the identification of SNP targets directly impacting gene expression and subsequent protein expression levels. Therefore, these could be ideal candidates for downstream functional studies using targeted knockouts and gene-editing studies. 

Our analysis and wide validation resulted in the identification of seven SNPs belonging to two genes. A C to T mutation at position 72672774 on chromosome 9 belongs to the gene (Solyc09g098310/LOC101258900), annotated as a putative cysteine-rich transmembrane (CYSTM) family protein. The role of CYSTMs has not been well understood and studies so far have indicated their role in plant stress responses by preventing ROS damage they are a component of the salicylic acid response pathway [19,20]. Some CYSTMs are induced by pathogens and hence are also known as pathogen-induced cysteine-rich transmembrane proteins (PCMs) and their overexpression helps in the defense against pathogens [20]. However, in our data, we see a downregulation of Solyc09g098310 in resistant plants upon infection, and upregulation in infected susceptible plants. The presence of a possible sly-miR395 interaction could be a key regulatory mechanism of Solyc09g098310 and needs further investigation. While CYSTMs contribute to various aspects of plant immunity against biotic and abiotic stress, one of the most important mechanisms for disease resistance against necrotic fungi is the use of the pathogen-associated molecular pattern (PAMP)-triggered immunity (PTI) [21]. These PAMPs are identified by pattern recognition receptors (PRRs), which trigger a cascade of reactions leading to a defense response. However, pathogens over time have developed a counter mechanism using virulence factors or effectors to evade PTI. This is typically countered by host plants using R (resistance) genes [21]. The study and use of R genes has been widely used as an important source of resistance in commercial tomato breeding [22]. We validated seven SNPs on Solyc09g098100/LOC101055591, an NLB-LRR domain-containing R gene on chromosome 9 and very close in proximity (~ 55 kb in SL3.0 genome) to the well-characterized late blight-resistant locus of *Ph-3* [15]. The putative role of Solyc09g098100 in the defense against *Phytophthora* was further strengthened by the interaction sites of sly-miR482d-3p [4,11], sly-miR5303 [23] and sly-miR6024 [24], which are well-known to have roles during *Phytophthora* and other fungal infections. The expression pattern showing upregulation of Solyc09g098100 and the presence of salicylic acid (SA) response element in the promoter region could indicate a salicylic acid-mediated response in the resistant plants. While multiple pieces of evidence in our study suggest the involvement of Solyc09g098100 in resistance, further studies are required to explore the role of the gene and associated polymorphisms to understand the underlying molecular mechanism and its regulation.

In conclusion, we identified and validated a set of SNPs in disease-responsive genes under *Phytophthora* infection in tomato plants. The selection of SNPs was achieved not only by comparing RNA-sequencing data of our plant material but also by integrating a large set of publicly available sequencing datasets with known phenotypes under *Phytophthora* infection. The association of the SNPs identified with maximal confidence was validated on additional plant material with contrasting phenotypes. The identified SNPs in this study are located on two pertinent genes (Solyc09g098310 and Solyc09g098100) with a putative role in plant defense. In addition to serving as routine assays for marker-assisted selection, these SNPs can also be functional targets for creating edited tomato varieties resistant to late blight. 

## 4. Materials and Methods

### 4.1. Plant Material, Fungal Infection, and Sampling

The plant material used in this study belongs to a collection of the Department of Agronomy, Food, Natural Resources, Animals and Environment (DAFNAE) at the University of Padova. Two tomato pre-breeding lines, one susceptible and one showing a high level of resistance against *Phytophthora infestans* were selected for this study. The resistant line was also characterized by the presence of *Ph-2* and *Ph-3* late blight-resistant loci. Selected resistant and susceptible lines were grown in pots under uniform conditions. Upon reaching 60 days, the leaves were sprayed with a suspension of *P. infestans* isolates from the Po Valley (Italy). Sporangia were collected from infected tomato leaves. To prepare a sporangial suspension for experimental inoculations, lesions were washed in 10 mL of sterile water. The concentration of the resulting sporangial suspension was determined by pipetting 4 µL of suspension onto a glass slide. The number of sporangia was calculated under a microscope. To achieve a final concentration of 10,000 sporangia mL^−1^, the suspension was diluted with sterile water and then chilled for 1 h at 4 °C before spraying. Inoculated and non-inoculated plants were kept in separate transparent boxes and placed in a controlled growth chamber with a 12 h light/dark cycle at 21 °C and 97–99% relative humidity.

For validation of identified markers, a total of 121 individuals including 14 resistant and 17 susceptible breeding lines with biological replicates were used.

### 4.2. Library Preparation and Transcriptome Sequencing

A total of 30 mg of fresh leaf samples were processed using Tissue Lyser (Qiagen, Germany) with 100 µL of the lysis-binding buffer. mRNAs were directly isolated using the Dynabeads mRNA Direct Micro Kit (Thermo Fisher Scientific, Carlsbad, CA, USA). The quality and quantity of isolated mRNAs were checked by Agilent TapeStation 1500 (Agilent Technologies Inc., Santa Clara, CA, USA). Sequencing libraries were prepared using Ion Total RNA-Seq Kit v2 (Thermo Fisher Scientific). The prepared libraries were loaded onto Ion 540 chip kit followed by sequencing using the Ion S5 GS system. 

### 4.3. Transcriptome Data Analysis

Demultiplexed raw sequences exported as unaligned BAM files from the Ion S5 GS sequencer were converted to fastq format using samtools 1.10 [25]. Quality check of the sequences was performed using FastQC v0.11.9 [26] and MultiQC Toolbox [27]. Quality-checked reads were aligned to Tomato (*Solanum lycopersicum* L.) reference genome SL3.0 (http://solgenomics.net (accessed on 29 December 2022)) using bowtie2 v2.3.5.1 [28]. Unaligned reads were then mapped against *Phytophthora infestans* (NCBI genome: ASM14294v1) to check for the presence of the fungal pathogen. Following this, the aligned files were converted into sorted and indexed BAM files using the samtools suite. Raw read counts for genes were calculated using samtools multiBamCov and further analyzed using Deseq2 [29] for library size normalization and differential gene expression. Principal component analysis was performed using the read counts matrix from DESeq2 and plotted using ggplot2 [30]. The normalized count matrix generated from DESeq2 was analyzed using the iDEP server [31] to explore gene ontology, functional clustering, and pathways analysis.

### 4.4. Variant Calling from Transcriptome Data and Integration of Publicly Available Data

Aligned bam files were processed using samtools mpileup [25] and bcftools call 1.13 [32] to call the variants against reference genome SL3.0 (http://solgenomics.net (accessed on 29 December 2022)). Variants were then filtered using the bcftools filter [32] with a cut-off of minimum QUAL (quality) of 10 and a minimum DP (depth) of 3. Next, the variants unique to resistant lines were screened based on multiple criteria. First, the variants coming from differentially expressed genes in any condition of comparison were selected. Following this, the variants belonging to genes already known to be involved in late blight resistance were screened based on a literature search and screening of the genome annotation file descriptions. Variants selected from the transcriptome data were checked with a list of publicly available known resistant and susceptible varieties and accession to build more confidence in discovered variants. The samples included a wide range of plants coming from commercial varieties of *S. lycopersicum* L. and accessions of *S. pimpinellifolium*. Raw fastq files were downloaded from NCBI-SRA (https://www.ncbi.nlm.nih.gov/sra) and processed using the same pipeline as done for transcriptome for variant calling. Datasets with available variant call files were directly compared. The list of public datasets used for the study is mentioned in Table 2.

### 4.5. Experimental Validation of SNPs

Variants selected after bioinformatics analysis were genotyped using rhAmp assays (Integrated DNA Technologies, USA) on a wide range of resistant and susceptible lines including 14 resistant and 17 susceptible lines with biological replicates. DNA from the samples was extracted and purified using BioSprint 96 workstation (QIAGEN, Germany) by the method described by Stevanato et al. 2015 [42]. Genotyping was performed in a 5 μL reaction volume with 5 ng of DNA, 2.65 μL of rhAmp Genotyping Master Mix, 0.25 μL of rhAmp SNP assay mix, and 1 μL of nuclease-free water with thermocycler conditions given in Broccanello et al. 2018 [43]. Allelic calls were performed using QuantStudio™ Design and Analysis software v1.4.3. Statistical significance was calculated by chi-square test using the web server icalcu.com/stat/chisqtest.html (accessed on 29 December 2022).

### 4.6. Characterization of Candidate Genes

Genes containing the validated SNPs of interest were first checked with ITAG 3.0 annotation (http:/solgenomics.net (accessed on 29 December 2022)) and subsequently with a newer version of *Solanum lycopersicum* L. genome annotation ITAG 4.0 (http://solgenomics.net (accessed on 29 December 2022)) for updated gene structure and annotation. NCBI protein BLAST [44] was used to annotate respective protein sequences against *Arabidopsis thaliana* and NCBI non-redundant database [45]. Further confirmation of annotation was done using domain identification using InterProScan classification [46]. PlantCARE database [47] was used for the identification of cis-regulatory sites for these genes using 1500 base pair upstream sequence and psRNATarget [48] was used to screen for miRNA interactions with the primary transcripts of these genes.

## Figures and Tables

**Figure 1 ijms-24-06798-f001:**
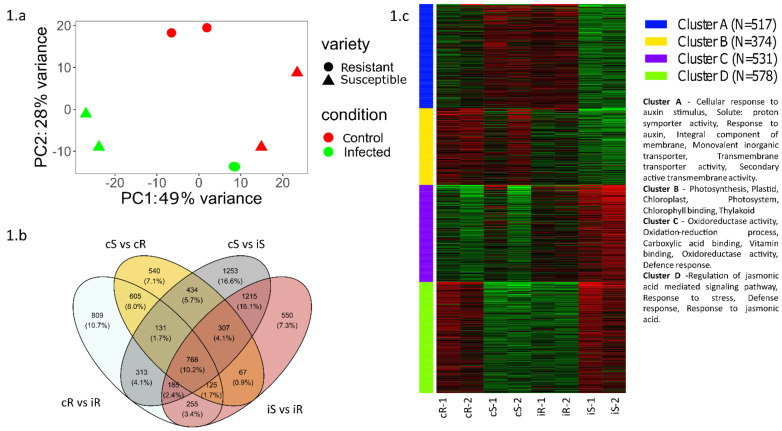
(**a**) Principal component analysis of samples from transcriptome analysis; (**b**) Venn diagram representing the differentially expressed genes in all conditions sequenced (cS−control susceptible, cR−control resistant, iS−infected susceptible, iR−infected resistant) and their overlap; (**c**) k−means clustering heatmap showing four distinct clusters of genes based on their expression and their gene ontology.

**Figure 2 ijms-24-06798-f002:**
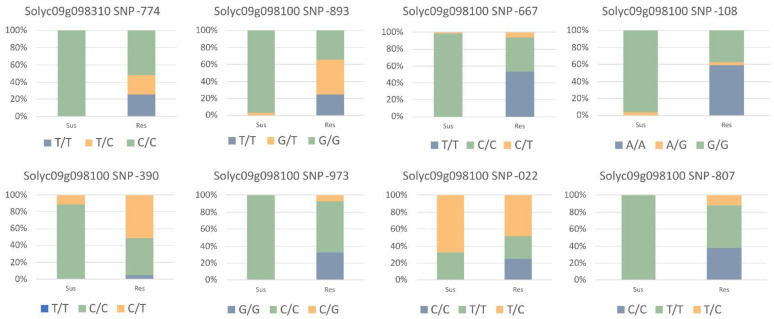
Genotype frequencies of validated SNP targets in a total of 121 individuals including 14 resistant and 17 susceptible breeding lines. The percentage of homozygous reference (blue), homozygous alternate (green), and heterozygous allele (orange) are shown in tested lines.

**Figure 3 ijms-24-06798-f003:**
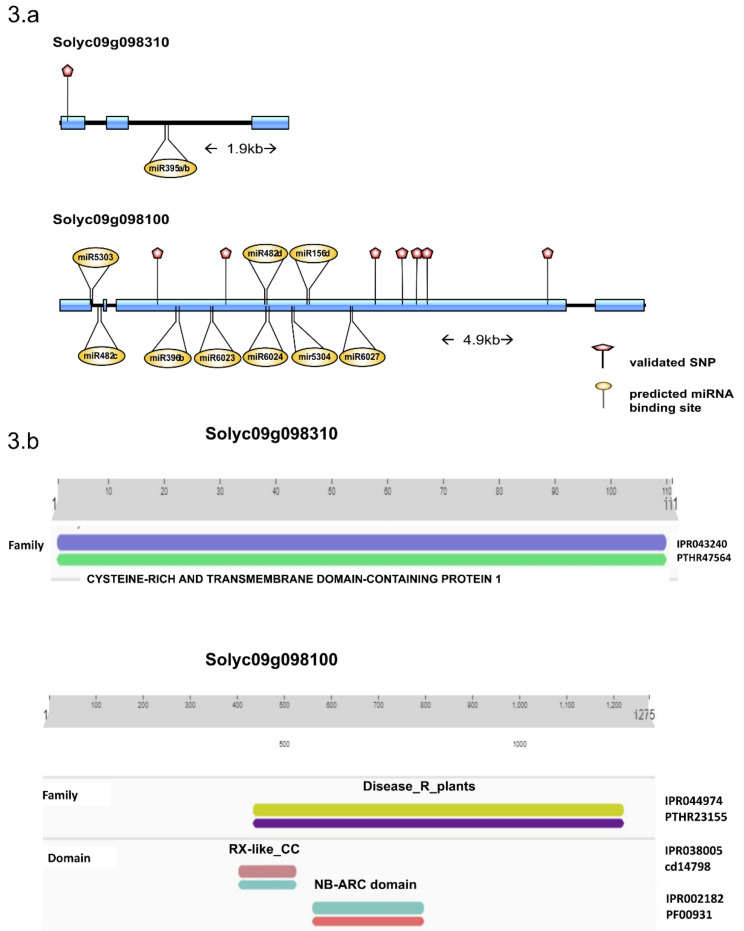
(**a**) Candidate gene structure of Solyc09g098310 and Solyc09g098100. The positions of validated SNPs within the exons (red pins) along with predicted miRNA binding sites (yellow pins) are presented. (**b**) The predicted protein domains Solyc09g098310 and Solyc09g098100 from InterProScan and their location is presented.

**Table 1 ijms-24-06798-t001:** Allele frequency distribution of validated SNP targets by rhAmp genotyping.

SNP ID	No. of Samples	Phenotype	Alternative Allele %	Reference Allele %	Chi-Squared	*p*-Value
774	65	Susceptible	0.00	100	45.4	*p* < 0.01
	73	Resistant	37.00	63		
893	67	Susceptible	1.50	98.5	53.4	*p* < 0.01
	73	Resistant	45.20	54.8		
667	52	Susceptible	1.00	99	75.8	*p* < 0.01
	52	Resistant	56.70	43.3		
390	53	Susceptible	5.70	94.3	20.5	*p* < 0.01
	43	Resistant	30.20	69.8		
973	53	Susceptible	0.00	100	44.4	*p* < 0.01
	55	Resistant	36.40	63.6		
022	70	Susceptible	33.60	66.4	4.8	*p* < 0.05
	81	Resistant	48.80	51.2		
807	55	Susceptible	0.00	100	56.5	*p* < 0.01
	42	Resistant	44.00	56		
108	53	Susceptible	1.90	98.1	80.5	*p* < 0.01
	56	Resistant	60.70	39.3		

**Table 2 ijms-24-06798-t002:** List of public datasets used in the study.

Variety	Literature Evidence Supporting the Phenotype of the Variety	Phenotype	Data Source
Stupice	Powell et al. 2014 [33]	Resistant	https://plantgarden.jp/ (accessed on 29 December 2022)
Brandywine Red	Gevens et al. 2013 [34]	Resistant	SRR5080039
Matt’s Wild Cherry	Gevens et al. 2013 [34]	Resistant	SRR5079877
Prudens Purple	Gevens et al. 2013 [34]	Resistant	SRR5080111
Legend	Gevens et al. 2013 [34]	Resistant	SRR5079916
Cherry Roma	Gevens et al. 2013 [34]	Resistant	SRR5080059
Green Zebra	Gevens et al. 2013 [34]	Resistant	SRR5080064
Mr. Stripey	Hansen et al. 2014 [35]	Resistant	SRR5080065
Lemon Drop	Hansen et al. 2014 [35]	Resistant	SRR5079871
Mexico Midget	James 2015 [36]	Resistant	SRR5080113
NC1-CELBR	Hansen et al. 2014 [35]	Resistant	https://solgenomics.net/ (accessed on 29 December 2022)
LA2093	Merk 2010 [37]	Resistant	SRR12039813
LA1673	Nowakowska et al. 2014 [38]	Resistant	DRR241605
San Marzano	Rodríguez et al. 2011 [39]	Susceptible	https://plantgarden.jp/ (accessed on 29 December 2022)
Castle rock	Arafa et al. 2017 [13]	Susceptible	https://plantgarden.jp/ (accessed on 29 December 2022)
Money Maker	Ojiewo et al. 2010 [40]	Susceptible	https://plantgarden.jp/ (accessed on 29 December 2022)
LA4084	Zhang et al. 2014 [41]	Susceptible	SRR1013253

## Data Availability

Transcriptome sequencing data can be found at ENA accession ID PRJEB59344.

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
