# Peer review of "Transcriptome-Assisted SNP Marker Discovery for Phytophthora infestans Resistance in Solanum lycopersicum L."

_ijms, 2023, doi:10.3390/ijms24076798_

Round 1

Reviewer 1 Report

Dear Authors,

I had a great opportunity to review manuscript entitled “Transcriptome-assisted SNP marker discovery for Phytophthora infestans resistance in Solanum lycopersicum L.” which is considered for publication in IJMS in MDPI Publishing. The article present new insight in investigation of resistance against Phytophthora infestans resistance in tomato plants. However article need improvments which list I present below:

Introduction section

This part according IJMS publication rules should ended with précised formulated aim or even aim and hypothesis of the study. Currently nothing like that is present in the manuscript and this must be updated.

Material and Methods section

ATTENTION: THIS PART according IJMS publication rules should be after Discussion section not after Introduction. It must be updated I suggest also to point by point to check all publication rules in IJMS by Authors because other problem with publication rules inadequate is also present. More serious problems the part “ 2.1. Plant material, fungal infection, and sampling” must provide exactly source of plant material and fungus isolates. Now any of this information is not present. The statement: “Upon reaching  60 days, the leaves were sprayed with P. infestans isolate mix” which isolates was in mix exactly this is crucial information method of spraying the elements of solution in which fungus was sprayed on plants. The authors did not provided the exact plant cultivation conditions before introduction of fungus and after treatment. All of this must be provided.

Results section

The some Figures is overloaded with data small and extremely low quality (bad pixels, unreadable even).  The examples is Figure 1 and 3. Problem the most extreme in Figure 1. Figure 1 must be divided to 3 separate figures. One wit maximum enlarged heat map and 2 nest figures with rest of the data

Sincerely,

Author Response

Point 1:

 Introduction section

This part according IJMS publication rules should ended with précised formulated aim or even aim and hypothesis of the study. Currently nothing like that is present in the manuscript and this must be updated.

Response: Thank you for the suggestion. We have updated the introduction as per your comments. The details are mentioned as follows:

“In this context, the aim of this study was to identify SNP markers in functional genes associated with disease resistance to Phytophthora infestans in tomato.  This was primarily achieved  using an  RNA sequencing-guided approach to understand the transcriptomic changes upon infection by Phytophthora infestans in resistant and susceptible breeding lines followed by the discovery of SNPs in disease-responsive genes dysregulated upon infection between resistant and susceptible lines. The selected SNP markers were additionally correlated with all publicly available sequencing datasets of tomato with contrasting disease phenotypes and finally validated on a test dataset of secondary lines with known disease phenotypes.”.

Point 2:

Material and Methods section

ATTENTION: THIS PART according IJMS publication rules should be after Discussion section not after Introduction. It must be updated I suggest also to point by point to check all publication rules in IJMS by Authors because other problem with publication rules inadequate is also present. More serious problems the part “ 2.1. Plant material, fungal infection, and sampling” must provide exactly source of plant material and fungus isolates. Now any of this information is not present. The statement: “Upon reaching  60 days, the leaves were sprayed with P. infestans isolate mix” which isolates was in mix exactly this is crucial information method of spraying the elements of solution in which fungus was sprayed on plants. The authors did not provided the exact plant cultivation conditions before introduction of fungus and after treatment. All of this must be provided.

Response: Thank you for the suggestion, we have moved the method section as per IJMS guidelines, and a more detailed paragraph on “plant material, fungal infection and sampling”  was added in the method section:

“The plant material used in this study belongs to the collection of the Department of Agronomy, Food, Natural Resources, Animals and Environment (DAFNAE), University of Padova. Two tomato pre-breeding lines, one susceptible and one showing a high level of resistance against Phytophthora infestans were selected for this study. The resistant line was also characterized by the presence of Ph-2 and Ph-3 late blight-resistant loci. Selected resistant and susceptible lines were grown in pots under uniform conditions. Upon reaching 60 days, the leaves were sprayed with a suspension of P. infestans isolates from Po valley (Italy). Sporangia were collected from infected tomato leaves. To prepare a sporangial suspension for experimental inoculations, lesions were washed in 10 mL of sterile water. The concentration of the resulting sporangial suspension was determined by pipetting 4 µL of suspension onto a glass slide. The number of sporangia was calculated under a microscope. To achieve a final concentration of 10,000 sporangia mL-1, the suspension was diluted with sterile water and then chilled for 1 h at 4°C before spraying.  Inoculated and non-inoculated plants were kept in separate transparent boxes and placed in a controlled growth chamber with a 12 h light/dark cycle at 21°C and 97–99% relative humidity.”

Point 3:

Results section

The some Figures is overloaded with data small and extremely low quality (bad pixels, unreadable even).  The examples is Figure 1 and 3. Problem the most extreme in Figure 1. Figure 1 must be divided to 3 separate figures. One wit maximum enlarged heat map and 2 nest figures with rest of the data

Response: Thank you for the suggestion. Figure qualities were improved, and updated with new enlarged images to make it clear.

Reviewer 2 Report

In this study, Deb et al. conducted a transcriptome study to assess the gene expression dynamics of resistant and susceptible tomato varieties to Phytophthora infection. They have identified and validated a panel of single nucleotide polymorphism (SNP) markers correlated with disease resistance. The paper is generally well-written and structured. I enjoyed reading the manuscript. However, minor edits should be addressed before the paper can progress further. I hope the authors find these comments helpful in maximizing the impact of their study.

1- I would suggest improving the quality of figures 1 and 2 to increase the readability.

2- In the Abstract: I would encourage the authors to rewrite the last sentence to be more specific, highlighting the study's main conclusion. 

3- Line 37" Leave space after "breeding" in "...and selective resistance breeding[2]"

4- Line 106: Leave space after "Ion" in "...unaligned BAM files from the IonS5 GS..."

5- Line 113: Fix the mistype after "suite" in "...dexed BAM files using samtools suite16."

6- Line 318: Remove the extra dot after "defense. ."

Author Response

Pont 1: 

I would suggest improving the quality of figures 1 and 2 to increase the readability.

Response: Thank you for the suggestion. Figure qualities were improved and updated with new enlarged images to make it clear.

Point 2:

In the Abstract: I would encourage the authors to rewrite the last sentence to be more specific, highlighting the study's main conclusion. 

Response- Thank you for the suggestion. We have added the study’s main conclusion in the abstract -

“The panel of SNPs generated in this study using transcriptome sequencing shows a correlation with disease resistance across a broad set of plant material which can be used for routine  molecular screening and is a valuable resource marker for the tomato breeding community. “

Point 3: 

Line 37" Leave space after "breeding" in "...and selective resistance breeding[2]"

Response: Thank you for the suggestion. We corrected them accordingly.

Point 4: 

Line 106: Leave space after "Ion" in "...unaligned BAM files from the IonS5 GS..."

Response: Thank you for the revision. We corrected them accordingly.

Point 5: 

Line 113: Fix the mistype after "suite" in "...dexed BAM files using samtools suite16."

Response: Thank you for the suggestion. We corrected them accordingly.

Point 6:

Line 318: Remove the extra dot after "defense. ."

Response: Thank you for the revision. We corrected them accordingly.